# A risk-adjusted and anatomically stratified cohort comparison study of open surgery, endovascular techniques and medical management for juxtarenal aortic aneurysms—the UK COMPlex AneurySm Study (UK-COMPASS): a study protocol

Shaneel R Patel [1,2,3] David C Ormesher,[1] Samuel R Smith,[4] Kitty H F Wong,[5] Paul Bevis,[5] Colin D Bicknell,[6] Jonathan R Boyle,[7] John A Brennan,[1] Bruce Campbell,[8] Andrew Cook [9] Alastair P Crosher,[10] Rui V Duarte [11] Murray M Flett,[12] Carrol Gamble,[13] Richard J Jackson,[13] Maciej T Juszczak,[14] Ian M Loftus,[15] Ian M Nordon,[16] Jai V Patel,[17] Kellie Platt,[13] Eftychia-Eirini Psarelli,[13] Peter C Rowlands,[10] John V Smyth,[18] Theodoros Spachos,[19] Leigh Taggart,[13] Claire Taylor,[13] Srinivasa Rao Vallabhaneni [1,2]

For numbered affiliations see end of article.

**Correspondence to**
Professor Srinivasa Rao Vallabhaneni;
fempop@liverpool.ac.uk

## ABSTRACT

**Introduction** In one-third of all abdominal aortic aneurysms (AAAs), the aneurysm neck is short (juxtarenal) or shows other adverse anatomical features rendering operations more complex, hazardous and expensive. Surgical options include open surgical repair and endovascular aneurysm repair (EVAR) techniques including fenestrated EVAR, EVAR with adjuncts (chimneys/endoanchors) and off-label standard EVAR. The aim of the UK COMPlex AneurySm Study (UK-COMPASS) is to answer the research question identified by the National Institute for Health Research Health Technology Assessment (NIHR HTA) Programme: 'What is the clinical and cost-effectiveness of strategies for the management of juxtarenal AAA, including fenestrated endovascular repair?'

**Methods and analysis** UK-COMPASS is a cohort study comparing clinical and cost-effectiveness of different strategies used to manage complex AAAs with stratification of physiological fitness and anatomical complexity, with statistical correction for baseline risk and indication biases. There are two data streams. First, a stream of routinely collected data from Hospital Episode Statistics and National Vascular Registry (NVR). Preoperative CT scans of all patients who underwent elective AAA repair in England between 1 November 2017 and 31 October 2019 are subjected to Corelab analysis to accurately identify and include every complex aneurysm treated. Second, a site-reported data stream regarding quality of life and treatment costs from prospectively recruited patients across England. Site recruitment also includes patients with complex aneurysms larger than 55 mm diameter in whom an operation is deferred (medical management). The primary outcome measure is perioperative all-cause mortality. Follow-up will be to a median of 5 years.

### Strengths and limitations of this study

► UK COMPlex AneurySm Study will include all patients undergoing repair of a complex abdominal aortic aneurysm (AAA) in England over a 24-month period in a comparative effectiveness analysis of high external validity.

► This study will involve the largest Corelab analysis of CT aortograms ever conducted to identify complex aneurysms according to predefined objective criteria pertaining to aneurysm neck anatomy.

► This study will capture every treatment method used to repair complex aneurysms.

► This study will incorporate methods of accounting for baseline risk and indication biases arising from physiological status and anatomical complexity, which will improve internal validity of comparative effectiveness analyses.

► The main limitation of the study design is a lack of random allocation, which leaves potential for unknown biases remaining uncorrected.

**Ethics and dissemination** The study has received full regulatory approvals from a Research Ethics Committee, the Confidentiality Advisory Group and the Health Research Authority. Data sharing agreements are in place with National Health Service Digital and the NVR. Dissemination will be via NIHR HTA reporting, peer-reviewed journals and conferences.

**Trial registration number** ISRCTN85731188.

BMJ

## INTRODUCTION

Approximately 4000 individuals undergo elective repair of an abdominal aortic aneurysm (AAA) each year in the UK[1] with the intention of preventing premature death from aneurysm rupture. The two distinct repair methods used, open surgical repair (OSR) and endovascular aneurysm repair (EVAR), each have unique advantages and disadvantages. EVAR is safer within the perioperative period, but less durable in the years to follow compared with OSR. Treatment costs also differ between the two methods.

The complexity of aneurysm repair depends mainly on anatomical detail relating to the segment of undilated aorta between renal arteries and the aneurysm, referred to as the aneurysm 'neck'. Ideal aneurysm neck has normal diameter in cross-section which is uniform along its length with little angulation, thrombus or calcification. Infrarenal aneurysms with at least 10–15 mm length of ideal aneurysm neck are amenable to OSR with aortic occlusion clamp below the renal arteries. Such anatomy also allows standard EVAR within 'Instructions for Use' (IFU) defined by manufacturers of stent-grafts implanted during EVAR. Some 40%–60% of aneurysms fall within this category[2–4] and there is a wealth of comparative effectiveness evidence relating to such patients.[5 6] The remainder, whose aneurysm neck is too short (commonly called 'juxtarenal') or otherwise unsuitable for standard EVAR within IFU (on-label device use), are pragmatically referred to as having a 'complex aneurysm'.

OSR of complex aneurysms involves aortic occlusion clamps at a variety of levels; below the renal arteries (infrarenal), below the superior mesenteric artery (suprarenal) or above the coeliac artery (supravisceral or supracoeliac) with increasing levels of physiological burden and technical difficulty as the clamp level moves higher. Complex aneurysms also call for advanced EVAR techniques such as fenestrated EVAR (FEVAR), and EVAR with adjuncts (including chimneys and endoanchors),[7] all of which are more complex than standard EVAR. Standard EVAR outside IFU (off-label standard EVAR) is also used in routine clinical practice to treat complex aneurysms.

Repair of complex aneurysms by any technique is associated with greater risk than repair of standard aneurysms. Therefore, in patients with aneurysms larger than the conventional treatment threshold of 55 mm diameter and poor physiological status, a shared decision may be made to defer operation until the risk versus benefit turns favourable towards intervention. Such 'medical management' of complex aneurysms with continued surveillance beyond the 55 mm size threshold has never been compared with immediate operation.

Existing evidence for different methods of repair of complex aortic aneurysms is largely in the form of case series. Most published analyses are limited by selection and indication biases. The practice of offering OSR to physiologically fitter patients and reserving EVAR techniques for the less fit patients is seldom accounted for in analyses comparing outcomes.[8–10] Furthermore, in most analyses, patients are grouped by the operation techniques used rather than the detail of the anatomy being treated. A lack of precise and objective information regarding aneurysm neck anatomy, pooling of short-neck aneurysms with aneurysms without a neck as well as aneurysms of the visceral aortic segment and thoracoabdominal aortic aneurysms seriously limits the value of most published evidence.[11–14] These sources of bias, confounding and heterogeneity, have been consistently recognised as weaknesses in the evidence base by authors of systematic reviews.[15 16]

There is therefore a pressing need to generate comparative effectiveness evidence in this area with adequate internal and external validity. Expert opinion sought by the study funder, as well as an independently conducted workshop revealed that in the UK, a randomised controlled trial comparing OSR against FEVAR (the two principal repair methods) would not be feasible due mainly to a lack of sufficient equipoise. Additionally, such a trial would neglect other methods of complex aneurysm repair that are widely used, including off-label EVAR and EVAR with adjuncts (such as endoanchors).

Endovascular treatments are associated with greater cost than OSR due to the cumulative costs of the stent-grafts, ancillary devices, postoperative surveillance and re-intervention. National Health Service (NHS) clinical service commissioners have noted an increasing demand for FEVAR in the absence of evidence regarding clinical effectiveness identifying which patients would benefit from this procedure and whether the substantial cost associated with the technique is an appropriate use of NHS resources. Prohibitive cost and lack of experience with FEVAR in many arterial centres have concomitantly led to a substantial use of off-label standard EVAR in complex neck anatomy, an intervention that is known to have poorer long-term results compared with on-label EVAR.[17] The National Institute for Health Research Health Technology Assessment (NIHR HTA) Programme has commissioned this comparative research study in order to resolve this NHS decision problem.

The primary aim of UK COMPlex AneurySm Study (UK-COMPASS) is to answer the research question identified by the NIHR HTA Programme: 'What is the clinical and cost-effectiveness of strategies for the management of juxtarenal AAA, including fenestrated endovascular repair?' The five objectives of this study are shown in table 1.

## METHODS AND ANALYSIS
### Overview

UK-COMPASS is an empirical cohort study comparing the clinical and cost-effectiveness of different management strategies used to repair complex aneurysms, incorporating statistical methods of correcting for confounding by physiological risk and anatomical complexity.

The study incorporates aspects which have been referred to by NIHR as 'efficient study design', where a

| **Table 1** | The UK-COMPASS objectives |
|---|---|
| Objective 1 | To compare different treatment strategies for their perioperative mortality and morbidity, corrected for confounding physiological and anatomical characteristics, in order to account for baseline risk and indication biases. |
| Objective 2 | To identify whether particular physiological and/or anatomical baseline characteristics are associated with better clinical outcomes or better health economic efficiency using one or other treatment strategies. |
| Objective 3 | To compare different treatment strategies in terms of overall survival and in terms of treatment failure in the long-term follow-up (stent-graft-related complications, secondary interventions, aneurysm-related mortality). |
| Objective 4 | To perform cost-effectiveness analyses from NHS and personal social care perspective to establish incremental cost-effectiveness ratios, comparing different treatment strategies in terms of cost per incremental gain in and quality-adjusted life years. |
| Objective 5 | To establish the clinical and cost utility of FEVAR and alternatives, in patients who are considered physiologically unfit for OSR, and to compare these against medical management. |

FEVAR, fenestrated endovascular aneurysm repair; NHS, National Health Service; OSR, open surgical repair; UK-COMPASS, UK COMPlex AneurySm Study.

substantial proportion of data required for the analysis is retrieved from routinely collected sources. This reduces the cost of conducting research, facilitates inclusion of a large number of patients within a short time frame and reflects 'real-world' practice. The study has two streams of data collection:

### Routinely collected data

This stream pertains to patients who have undergone repair of a complex aneurysm in England, over a 2-year period between 1 November 2017 and 31 October 2019. Duration of inclusion period was estimated to be 2 years commencing as soon as regulatory approvals are in place, anticipating at least 30% of aneurysms to fulfil inclusion criteria in order to provide adequate statistical precision. Health data, demographic data and healthcare resource use data relating to operated patients will be retrieved from Hospital Episode Statistics (HES) datasets (NHS Digital), the National Vascular Registry (NVR), and the NHS Picture Archiving and Communication System (PACS). Median follow-up will be 5 years, until 31 October 2023.

The NVR is run by the Clinical Effectiveness Unit of the Royal College of Surgeons of England as a national clinical audit. Audit of NVR data revealed 95% case ascertainment and data completeness between 85.5% and 100% depending on the data point, with a median completeness of 93.3%. Data capture system incorporates a number of validations ensuring high level of accuracy.[1] The NVR includes a wide range of data relevant to this study, including demographics, comorbidity, preoperative assessment, intraoperative detail, postoperative complications, duration of hospital stay and critical care use for all types of AAA repair.

HES is a data warehouse containing details of all admissions, outpatient appointments, investigations and emergency department attendances at NHS hospitals in England. These data are collected during a patient's time at hospital and are designed to enable administration as well as medical research. In addition to health and resource use data during the primary admission for aneurysm repair, follow-up data will be collected for 5 years after the operation to identify late complications, treatment of complications and re-interventions. Survival data are retrieved via Office for National Statistics data linked to HES. The quality of HES data is recognised to be adequate for research and is regularly used for research purposes.[18]

### Site-reported data

► Quality of life (QoL) data prospectively reported by consenting patients (both operated and medically managed) from participating arterial centres in England will supplement the routinely collected data. Recruitment will be undertaken between 1 November 2017 and 31 December 2022, with follow-up until 31 December 2023.
► Baseline clinical data: participating centres will report baseline health and demographic data for patients recruited to the QoL aspect of the study.

Primary outcome measures are perioperative mortality, late all-cause mortality and late aneurysm-related mortality.

Secondary outcome measures are perioperative complication rates (including organ system complications, infections and complications specific to stent-grafts), perioperative secondary interventions, length of stay, intensive care usage, late complication rates (including complications specific to stent-grafts, renal failure/dialysis, aneurysm complications, infection, incisional hernia), late secondary intervention and health economic measures including incremental cost-effectiveness ratio in terms of cost per incremental gain in quality-adjusted life years.

All primary and secondary outcome measures except the health economic measures will be derived solely from the routinely collected data sources (HES, NVR and imaging). The health economic outcomes will be derived from a combination of routinely collected data sources (HES) and site-reported data sources (QoL collection).

All interventions that have been used in the study population will be compared as numbers allow. We anticipate methods of repair to include open surgery (OSR) with a variety of clamp levels (infrarenal, suprarenal or supravisceral), FEVAR, off-label use of standard EVAR and EVAR with adjuncts (including chimneys and endoanchors).

## METHODS
### Routinely collected data
The study population of complex aneurysms for the routinely collected data stream will be identified through anonymised Corelab analysis of preoperative CT scans; a Corelab refers to a combination of infrastructure and methodology of image analysis according to predetermined reporting standards and definitions. Corelab CT analysis will be undertaken for all AAAs repaired in England between 1 November 2017 and 31 October 2019. These patients and details regarding their preoperative CT scans will be identified from the HES dataset and Diagnostic Imaging Dataset of NHS Digital. This will permit retrieval of CT scans from the provider site to the Corelab based at the Royal Liverpool University Hospital using the Internet Exchange Portal and PACS.

The CT scans will be analysed in the Corelab using a prespecified protocol that was devised using a clinical consensus exercise conducted with experts from throughout the UK in December 2019. The Corelab protocol is provided in full in the online supplemental file and includes information on anatomical definitions, in-depth information regarding the measurement process, as well as the results of a validation exercise. The anatomical CT-based inclusion and exclusion criteria for the routinely collected data stream are summarised in table 2.

It is anticipated that approximately 8000 preoperative CT scans will be analysed in the Corelab and that of these, approximately 3000 patients will meet the anatomical inclusion criteria for a complex neck. These patients are included in the analysis without further selection or randomisation. Pseudonymised baseline and postoperative follow-up data (up to 5 years) will be provided for this population of patients with complex necks, using datasets from NHS Digital and the NVR. Additionally, intraoperative images and postoperative surveillance for the patients treated by endovascular techniques will be retrieved to the study Corelab.

### Site-reported data
All arterial vascular centres in England are eligible to participate as recruitment centres for the site-reported data stream. Recruitment will be undertaken between 1 November 2017 and 31 December 2022, with follow-up QoL data collected until 31 December 2023.

Patients will be prospectively identified at recruiting sites and approached for QoL follow-up using Good Clinical Practice-compliant processes. The anatomical inclusion criteria used to identify the patients mirror those of routinely collected data (see table 2). Patients will be eligible for contribution to QoL data collection if they are due to have a complex aneurysm managed with open or endovascular surgery, as well as cases that have reached 55 mm in maximal diameter and are selected to remain on extended surveillance ('medical management') with decision for treatment made at a later date once the aneurysm has reached a larger threshold diameter.

In addition to the anatomical exclusion criteria listed in table 2, cases using surgeon-modified stent-grafts and cases in which patients are outright turned down for surgery due to comorbidity or patient choice ('operation declined') will be ineligible for recruitment to QoL follow-up.

Both generic (EuroQuol-5D-5L, EQ-5D-5L[19]) and aneurysm-specific (Aneurysm-Dependent Quality of Life, Aneurysm DQoL[20]) QoL questionnaires will be used to collect data according to prespecified time points for both operated and medically managed patients as shown in table 3. The data will be collected in the form of written completed questionnaires or telephone interviews, in accordance with patient preference with raw data entered

| Table 2 | CT-based anatomical inclusion and exclusion criteria for the routinely collected data stream | |
|---|---|
| **Inclusion criteria** | **Exclusion criteria** |
| Aneurysm diameter ≥55 mm AND neck length <10 mm | Ruptured aneurysm |
| Aneurysm diameter ≥55 mm AND neck length ≥10 mm AND the presence of ≥1 of the following adverse neck features:<br>► Beta (β) angle >60°<br>► Conicality (>10% change in diameter along 15 mm length of neck)<br>► Thrombus lining >1/3 circumference of the neck OR filling 1/3 the surface area of the neck along a 3 mm length of neck<br>► Calcium load in the wall of the neck affecting >1/3 circumference of the neck along a 3 mm length of neck<br>► Non-straight neck, defined by the presence of intraneck angulation of >60° with a juxtarenal neck length of <15 mm<br>► Alpha (α) angle >45° | Subthreshold aneurysm (<55 m)<br><br>Thoracic aortic aneurysm<br><br>Thoracoabdominal aortic aneurysm<br><br>Visceral aortic aneurysm (where aorta at level of SMA is ≥30 mm) |

SMA, superior mesenteric artery.

**Table 3** Quality of life (QoL) data collection schedule

| QoL tool | Preop | 1 month postop | 3 months postop | 6 months postop | 1 year postop | Annually up to 5 years postop |
|---|---|---|---|---|---|---|
| Operated patients | | | | | | |
| EQ-5D-5L | ✓ | ✓ | | ✓ | ✓ | ✓ |
| Aneurysm-DQoL | ✓ | | ✓ | | ✓ | ✓ |
| Health diary | | | | | ✓ | ✓ |
| Medically managed patients (time points relative to recruitment/baseline questionnaire date and not the date of an operation) | | | | | | |
| EQ-5D-5L | ✓ | | | ✓ | ✓ | ✓ |
| Aneurysm-DQoL | ✓ | | | | ✓ | ✓ |
| Health diary | | | | | ✓ | ✓ |

Aneurysm-DQoL, Aneurysm-Dependent Quality of Life; EQ-5D-5L, EuroQuol-5 Dimension-5 Level.

and stored onto secure database by trained staff at the Liverpool Clinical Trials Centre. Change in overall QoL between baseline and subsequent intervals will be analysed for statistical significance. This information will also inform health economic analyses. A patient diary will also be completed by patients annually to provide data on health resource use following their aneurysm treatment.

Recruiting sites will also be asked to conduct a microcosting exercise in order to assess variations in endovascular consumables and local practice. A detailed cost profile will be obtained for at least 10 patients treated by each treatment modality: OSR, FEVAR and EVAR (±adjuncts). A proforma for microcosting data collection is included in the online supplemental file.

## Sample size calculations
### Perioperative mortality
The primary outcome measure is perioperative death. The primary efficacy parameter is the OR comparing FEVAR with both open repair and off-label EVAR. For OSR, conservative estimate of 8% perioperative death might be expected, and a reduction to 4% could be considered clinically relevant, which is equivalent to an OR of 0.48 (based on NVR 2015 report[21] and GLOBALSTAR registry[22]). This effect is slightly larger than that noted in randomised controlled trials of standard infrarenal aneurysm repair.[5] A Bonferroni adjusted two-sided alpha level of 0.025 is used. Based on these assumptions, collecting data on 2000 patients will ensure a power in excess of 80% if the allocation between treatments is relatively equal and should be preserved above 70% even if one treatment strategy is used twice as much as the other two.

### Stratified comparative analysis for perioperative mortality and identification of risk factors
Based on the best available estimates, it is anticipated that 120 events (perioperative deaths) will be available if 2000 patients are included in the study. The aim is to identify anatomical/physiological characteristics for which there is a differential effect on patient outcome depending on the treatment strategy employed. We plan

to use multivariable models which include propensity scores as continuous covariates, adjusted for appropriate confounding variables and including interaction effects with treatment effect. Using the statistical rule of thumb of 10 events per variable, approximately 12 variables can be included in any single model. If, for example, a two-level factor is to be investigated, main effects and interaction effects and a covariate of propensity scores will account for six variables, allowing for a further inclusion of other confounding variables as required.

### Long-term survival; all-cause mortality
Estimated survival rates at 1, 2 and 4 years are 90%, 80% and 70%, respectively. All patients will have a median of 5-year follow-up after entering the study. It is approximated that 2000 patients should then contribute approximately 550 events (all-cause mortality after 30 days/ discharge). As the study is not designed to identify a difference in long-term overall survival, no power calculation is provided to detect some minimum clinically relevant difference. Instead, assuming the difference between two treatment arms to be measured using a log HR, it is estimated that from 550 events, an SE of approximately 0.085 will be observed. This translates to a 95% CI of length of approximately 0.34. Here, an HR smaller than 0.71 or larger than 1.40 will be shown to be significant at a 5% level.

### Quality of life
We will obtain QoL data from a representative sample of patients undergoing AAA surgery during the study period. This will equate to 800 patients. We expect a 50% loss to follow-up, through non-compliance and natural attrition from death during the follow-up period. We plan to obtain QoL from 300 medically managed patients. Three hundred patients will give a 95% CI of maximum width 12.5 for the difference in mean EQ-5D-5L Visual Analogue Scale (EQ-VAS) values between baseline and later time points based on an SD equal to 20. The SD was calculated after an evaluation of EQ-VAS scores across a wide range of trials.

## Data analysis plan

Analyses will be carried out on an intention-to-treat basis, retaining all patients in the surgery group to which they are assigned irrespective of any procedural issues. The intervention groups will be OSR, FEVAR, EVAR±adjuncts (including chimneys and endoanchors). The exact treatment received will be determined by triangulation of NVR data and HES data for OSR. Patients receiving endovascular treatment will be identified from NVR data and HES data, and exact technical details of each procedure (including the use of adjuncts) will be determined through analysis of procedural images in the Corelab.

As an analysis of observational cohort data, all attempts to present a measure of efficacy free from confounding bias will be undertaken. Patients will be stratified by infrarenal neck length for analysis into three groups: 0-4 mm (juxtarenal), 5–9 mm (juxtarenal) and ≥10 mm neck length (not juxtarenal but complex based on device-specific IFU criteria). This will serve to limit anatomical heterogeneity and bias within comparisons. Additionally, patients will be stratified by physiological risk. The British Aneurysm Repair Score[23] will be used (11 clinical parameters available from NVR data) to divide each comparison group into those that are high risk (the highest risk quartile) and the remainder as standard risk.

Statistical analyses of the primary outcome shall follow the principle of propensity score analysis. Propensity score estimation is chosen for the analysis of the primary endpoint instead of multivariable regression techniques because it is considered the most appropriate means of accounting for selection bias at the low level of perioperative death rates anticipated.[24] Propensity scores estimate the probability of a patient being given each treatment strategy based on baseline clinical and demographic characteristics. Patients can then be stratified based on their propensity to receive each treatment strategy and an estimate of treatment effect is then obtained, which adjusts for potential selection bias in the data.

Models to estimate the propensity for each patient shall be generated using multivariable logistic modelling techniques using as candidate covariates only variables that influence simultaneously the treatment assignment and the outcome variable. Covariates shall be retracted to demographic/clinical data available at the point of treatment decision with the aim of producing a propensity model that satisfies the assumptions of conditional independence. The final analysis shall be carried out using stratified logistic where the strata are defined by the quintiles of the fitted propensity score values. Prior to analyses, it shall be confirmed that each stratum contains at least 20 observed perioperative deaths. If not, the strata will be combined with an adjacent strata level.

The key efficacy parameter of interest will be an OR presented with a 97.5% CI. Results will be presented as a forest plot with the OR within each stratum presented along with the overall effect.

Reports of HES data validation confirm high levels of completeness and accuracy. Therefore, missing data or inaccurate data are unlikely to be a significant problem. It is proposed that covariates with small amounts of missing data (≤10%) will be included on a complete case basis and covariates with a large amount of missing data (>10%) will be excluded from the analysis.

Sensitivity analyses are carried out to ensure the results are not sensitive to either the propensity model or the method of matching. With respect to the propensity model, sensitivity analyses shall be carried out by removing each covariate included in calculating the propensity score and repeating the analysis of the primary endpoints. The results of all sensitivity analyses shall be presented in a forest plot along with the overall effect from the primary analysis.

It is recognised that recruitment for site-reported QoL data will occur either side of a cessation of elective services in the UK due to the COVID-19 pandemic in 2020/2021. There may be differences in patient behaviour and clinical practice before and after this period and we aim to investigate this further in the analysis phase. First, outcome data before and after the dates of interest will be compared. Second, clinical practice before and after the dates of interest will be compared using data from the NVR.

## Health economic analysis

The UK-COMPASS will adapt the recent National Institute for Health and Care Excellence (NICE) health economic analysis model[25] which will facilitate updates of NICE Clinical Guideline 'Abdominal Aortic Aneurysm–Diagnosis and Management (NG156)'.

Our aim is to inform NICE decision-making in a timely manner. The NICE health economic model has been developed from a UK NHS and Personal Social Services perspective, appraised by a NICE committee and scrutinised during consultation, and we see this fit to be adapted without developing a de novo model.

The purpose of the health economic analysis in this study is defined in the original objectives 4 and 5. The main outputs therefore will remain (a) estimation of incremental cost-effectiveness ratios comparing different treatment strategies in terms of life years and quality-adjusted life years gained, and (b) to identify whether particular level of physiological fitness (stratified into two groups) and/or anatomical baseline characteristics (stratified into three groups) are associated with better health economic efficiency using one or other treatment strategies.

At least 5-year follow-up is considered the essential minimum to capture attrition of treatment utility and accrual of treatment costs that occur from late failure/secondary intervention and postoperative surveillance.

## Patient and public involvement

The Liverpool Aneurysm Research patient and public involvement (PPI) group has been actively involved in the study. The study proposal was initially presented to the group on 2 November 2015 with subsequent follow-up.

Our PPI representatives strongly welcomed the proposed research and supported the team throughout the process of grant application and subsequently in the oversight of the study. The PPI group considered the dilemmas arising out of access to routinely collected data for this research project, including NHS number and date of birth, without explicit and specific consent from each patient. They scrutinised and approved the data management plan.

With appropriate safeguards in place, they felt that research using routinely collected data in the manner proposed here may have some advantages, by potentially reducing anxiety associated with being approached to participate in a prospective clinical study. They have also pointed out the benefits of data collected independent of physician or patient preference for a research study, thereby reducing bias. They welcomed the benefits of improved efficiency and improved generalisability of results across wider NHS.

The PPI group has helped us review the lay summary and has also reviewed the grant application at all stages. Once the study has commenced, the PPI group has been expended and annual meetings are expected to monitor progress. The Trial Steering Committee has one PPI member as an independent member. The PPI group is also expected to help with dissemination of research findings by reviewing patient education material as well as supporting commissioning decisions as appropriate.

## ETHICS AND DISSEMINATION

The study protocol, informed consent form, patient information sheet and all other patient-facing study documentation have received the favourable opinion of the North West–Preston Research Ethics Committee and Health Research Authority (HRA) approval.

Access to patient data in NHS Digital datasets, the NVR dataset as well as to imaging, without the need for patient consent, has been granted by the Confidentiality Advisory Group (CAG); CAG is an independent body which provides expert advice to the HRA and the Secretary of State for Health on whether applications to access confidential patient or service user information without consent should or should not be approved.

The study is managed by the Liverpool Clinical Trials Centre. The sponsor for the study is Liverpool University Hospitals NHS Foundation Trust.

### Dissemination plan

The results of this study will be written up and published as an official report for the NIHR HTA. It will also be published in a peer-reviewed journal and presented at international conferences.

### Author affiliations

[1]Vascular Surgery, Liverpool University Hospitals NHS Foundation Trust, Liverpool, UK
[2]Liverpool Centre for Cardiovascular Science, University of Liverpool, Liverpool, UK
[3]Cardiovascular and Metabolic Medicine, University of Liverpool, Liverpool, UK
[4]School of Medicine, University of Liverpool, Liverpool, UK
[5]Vascular Surgery, North Bristol NHS Trust, Bristol, UK
[6]Department of Surgery and Cancer, Imperial College London, London, UK
[7]Vascular Surgery, Cambridge University Hospitals NHS Foundation Trust, Cambridge, UK
[8]Vascular Surgery, Royal Devon and Exeter Hospital, Exeter, UK
[9]Wessex Institute, University of Southampton, Southampton, UK
[10]Radiology, Liverpool University Hospitals NHS Foundation Trust, Liverpool, UK
[11]Liverpool Reviews and Implementation Group, University of Liverpool, Liverpool, UK
[12]Vascular Surgery, Ninewells Hospital, Dundee, UK
[13]Liverpool Clinical Trials Centre, University of Liverpool, Liverpool, UK
[14]Vascular Surgery, University Hospitals Birmingham NHS Foundation Trust, Birmingham, UK
[15]Vascular Surgery, St George's University Hospitals NHS Foundation Trust, London, UK
[16]Vascular Surgery, University Hospital Southampton NHS Foundation Trust, Southampton, UK
[17]Interventional Radiology, Leeds Teaching Hospitals NHS Trust, Leeds, UK
[18]Vascular Surgery, Manchester University NHS Foundation Trust, Manchester, UK
[19]Vascular Surgery, Lancashire Teaching Hospitals NHS Foundation Trust, Preston, UK

**Acknowledgements** We would like to acknowledge the following UK-COMPASS committees and advisors: Trial Steering Committee (TSC) Janet Powell (Independent chair), Florian Dick (Independent specialist), Jonathan Michaels (Independent specialist, Health Economics), Andrew Tambyraja (Independent specialist) and Paul Massey (Public and Patient Involvement representative). Independent Data Monitoring and Safety Committee (IDMSC): Mike Roberts (Chair), Viktoria McMillan, Ulrike Naumann and Murray Flett.Trial Management Group (TMG): Barbara Arch, Colin Bicknell, James Boateng, Jonathan Boyle, Bruce Campbell, Andrew Cook, Alastair Crosher, Rui Duarte, Carol Gamble, Richard Jackson, David Ormesher, Shaneel Patel, Kellie Platt, Heather Rogers, Leigh Taggart, Claire Taylor and Syed Yusuf. Patient advisers: Paul Massey as a Public and Patient Involvement representative for the study

**Contributors** Roles of each author are as follows: SRP—study design, planning, conduct, data acquisition, interpretation of data, writing of the manuscript and editing of the manuscript. DCO—study design, planning, conduct, data acquisition, interpretation of data, writing of the manuscript and editing of the manuscript. SRS—study conduct, data acquisition, interpretation of data and editing of the manuscript. KHFW—study planning, writing of the manuscript and editing of the manuscript. PB—study design, planning and editing of the manuscript. CB—study design, planning and editing of the manuscript. JRB—study design, planning and editing of the manuscript. JAB—study design, planning and editing of the manuscript. BC—study design, planning and editing of the manuscript. AC—study design, planning and editing of the manuscript. AC—study conduct, data acquisition, interpretation of data and editing of the manuscript. RD—study design, planning and editing of the manuscript. MMF—study design, planning, interpretation of data and editing of the manuscript. CG—study design, planning, interpretation of data and editing of the manuscript. RJJ—study design, planning, acquisition of data, interpretation of data and editing of the manuscript. MTJ—study design, planning and editing of the manuscript. IL—study design, planning and editing of the manuscript. IMN—study design, planning and editing of the manuscript. JP—study design, planning and editing of the manuscript. KP—study conduct, planning and editing of the manuscript. EEP—study design, planning, acquisition of data, interpretation of data and editing of the manuscript. PCR—study design, planning, acquisition of data and editing of the manuscript. JVS—study design, planning and editing of the manuscript. TS—study design, planning and editing of the manuscript. LT—study conduct, planning and editing of the manuscript. CT—study conduct, planning and editing of the manuscript. SRV—chief investigator, study design, planning, conduct, data acquisition, interpretation of data, writing of the manuscript and editing of the manuscript.

**Funding** This work was solely funded by the National Institute for Health Research Health Technology Assessment (NIHR HTA) Programme (grant award ID: 15/153/02).

**Competing interests** None declared.

**Patient consent for publication** Not required.

**Provenance and peer review** Not commissioned; externally peer reviewed.

**Open access** This is an open access article distributed in accordance with the Creative Commons Attribution 4.0 Unported (CC BY 4.0) license, which permits others to copy, redistribute, remix, transform and build upon this work for any purpose, provided the original work is properly cited, a link to the licence is given, and indication of whether changes were made. See: https://creativecommons.org/licenses/by/4.0/.

**ORCID iDs**
Shaneel R Patel http://orcid.org/0000-0002-4388-2619
Andrew Cook http://orcid.org/0000-0002-6680-439X
Rui V Duarte http://orcid.org/0000-0001-6485-7415
Srinivasa Rao Vallabhaneni http://orcid.org/0000-0002-4362-9277

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
