## [Reviewer comments · BMJ Open]

ARTICLE DETAILS

TITLE (PROVISIONAL)	A risk-adjusted and anatomically stratified cohort comparison study of open surgery, endovascular techniques and medical management for juxtarenal aortic aneurysms: The UK COMpLex Aneurysm Study (UK-COMPASS) - A Study Protocol.
AUTHORS	Patel, Shaneel; Ormsher, David; Smith, Samuel; Wong, Kitty; Bevis, Paul; Bicknell, Colin; Boyle, Jonathan; Brennan, John; Campbell, Bruce; Cook, Andrew; Crosher, Alastair; Duarte, Rui; Flett, Murray; Gamble, Carrol; Jackson, Richard; Juszczak, Maciej; Loftus, Ian; Nordon, Ian; Patel, Jai; Platt, Kellie; Psarelli, Eftychia; Rowlands, Peter; Smyth, John; Spachos, Theodoros; Taggart, Leigh; Taylor, Claire; Vallabhaneni, Srinivasa

VERSION 1 – REVIEW

REVIEWER	Hamming, Jaap LUMC, surgery
REVIEW RETURNED	28-Jun-2021

GENERAL COMMENTS	This is an intensive study on aneurysm details and compare them with the outcome. The possibilities of extensive review of the CTA data will lead to good insight in indications and possible problems with certain anatomical aspects. The are some questions: - It is somewhat difficult to understand what data are retrieved exactly from which data base: long term survival? QoL? maybe table would be helpful for this- how complete and accurate are all the databases? Were there audits performed to evaluate the reliability of the data?- why the limitation of the period jan 2017-oct 2019? why not earlier, why not extending?- How were the QoL data obtained?: did all patients fill in the questionnaires even preoperatively? in which database were they filed? houw and by whom will they be interpreted?
---

REVIEWER	Milner, Ross University of Chicago
REVIEW RETURNED	01-Aug-2021

GENERAL COMMENTS	Thank you for your well written paper. One minor comment: Did you mean to say this?: "OSR is safer within the perioperative period, but more durable in the years to follow compared to EVAR."
--

VERSION 1 – AUTHOR RESPONSE

Reviewer: 1

Dr. Jaap Hamming, LUMC

Comments to the Author:

This is an intensive study on aneurysm details and compare them with the outcome. The possibilities of extensive review of the CTA data will lead to good insight in indications and possible problems with certain anatomical aspects.

There are some questions:

It is somewhat difficult to understand what data are retrieved exactly from which data base: long term survival? QoL? maybe table would be helpful for this

The data sources are described in the Routinely collected data and Site reported data subsections on Pages 6 and 7. To further clarify how these relate to the outcome measures, the following text has been added to page 7:

“All primary and secondary outcomes measures except the health economic measures will derive solely from the routinely collected data sources (HES, NVR and Imaging). The health economic outcomes will derive from a combination of routinely collected data sources (HES) and site-reported data sources (QOL collection).”

How complete and accurate are all the databases? Were there audits performed to evaluate the reliability of the data?

We have added the following information to the manuscript regarding completeness, accuracy, and audit of the data sources:

Regarding NVR, added to page 6:

Audit of NVR data revealed 95% case ascertainment and data completeness between 85.5% and 100% depending upon the data point, with a median completeness of 93.3%. Data completeness Data capture system incorporates a number of validations ensuring high level of accuracy.

(<https://www.vsqip.org.uk/content/uploads/2020/11/NVR-2020-Annual-Report.pdf>)

Regarding HES, added to page 6:

The quality of HES data is recognised to be adequate for research and is regularly used for research purpose. (Herbert A, Wijlaars L, Zylbersztejn A, Cromwell D, Hardelid P. Data Resource Profile: Hospital Episode Statistics Admitted Patient Care (HES APC). *Int J Epidemiol.* 2017;46(4):1093-1093i)

Why the limitation of the period jan 2017-oct 2019? why not earlier, why not extending?

Duration of inclusion period was estimated to be 2 years commencing as soon as regulatory approvals are in place, anticipating at least 30% of aneurysms to fulfil inclusion criteria in order to provide adequate statistical precision. We have added this information to the manuscript on page 6.

How were the QoL data obtained? did all patients fill in the questionnaires even preoperatively? in which database were they filed? how and by whom will they be interpreted?

The QOL data collection process is described on Page 8. We describe the specific questionnaires used and the methods of collection. The time points are shown in Table 3 (including the pre-operative baseline questionnaire). We have added the following text on Page 8 to clarify the process:

“The data will be collected in the form of written completed questionnaires or telephone interviews, in accordance with patient preference with raw data entered and stored onto secure database by trained staff at the Liverpool Clinical Trials Centre. Change in overall QoL between baseline and subsequent intervals will be analysed for statistical significance. This information will also inform health economic analyses”.

Reviewer: 2

Dr. Ross Milner, University of Chicago

Comments to the Author:

Thank you for your well written paper. One minor comment:

Did you mean to say this?: "OSR is safer within the perioperative period, but more durable in the years to follow compared to EVAR."

Thank you for highlighting this error. We have amended the sentence and it now reads as follows:

"EVAR is safer within the perioperative period, but less durable in the years to follow compared to OSR".